# Marine Origin Ligands of Nicotinic Receptors: Low Molecular Compounds, Peptides and Proteins for Fundamental Research and Practical Applications

**DOI:** 10.3390/biom12020189

**Published:** 2022-01-23

**Authors:** Igor Kasheverov, Denis Kudryavtsev, Irina Shelukhina, Georgy Nikolaev, Yuri Utkin, Victor Tsetlin

**Affiliations:** 1Shemyakin-Ovchinnikov Institute of Bioorganic Chemistry, Russian Academy of Sciences, 117997 Moscow, Russia; iekash@ibch.ru (I.K.); kudryavtsev@ibch.ru (D.K.); ner-neri@yandex.ru (I.S.); utkin@ibch.ru (Y.U.); 2Department of Biology, Moscow State University, 119234 Moscow, Russia; humanoid5@yandex.ru

**Keywords:** nicotinic acetylcholine receptors, ligands, marine alkaloids, conopeptides, snake neurotoxins, acetylcholine-binding proteins, extracellular domains

## Abstract

The purpose of our review is to briefly show what different compounds of marine origin, from low molecular weight ones to peptides and proteins, offer for understanding the structure and mechanism of action of nicotinic acetylcholine receptors (nAChRs) and for finding novel drugs to combat the diseases where nAChRs may be involved. The importance of the mentioned classes of ligands has changed with time; a protein from the marine snake venom was the first excellent tool to characterize the muscle-type nAChRs from the electric ray, while at present, muscle and α7 receptors are labeled with the radioactive or fluorescent derivatives prepared from α-bungarotoxin isolated from the many-banded krait. The most sophisticated instruments to distinguish muscle from neuronal nAChRs, and especially distinct subtypes within the latter, are α-conotoxins. Such information is crucial for fundamental studies on the nAChR revealing the properties of their orthosteric and allosteric binding sites and mechanisms of the channel opening and closure. Similar data are provided by low-molecular weight compounds of marine origin, but here the main purpose is drug design. In our review we tried to show what has been obtained in the last decade when the listed classes of compounds were used in the nAChR research, applying computer modeling, synthetic analogues and receptor mutants, X-ray and electron-microscopy analyses of complexes with the nAChRs, and their models which are acetylcholine-binding proteins and heterologously-expressed ligand-binding domains.

## 1. Introduction

Nicotinic acetylcholine receptors (nAChRs) were the first receptors/ion channels characterized in structural and functional detail, and the important role of low-molecular weight ligands (nicotine, acetylcholine) is reflected in the receptor name, while protein neurotoxins from snake venoms (as α-neurotoxins) played a crucial role in the first nAChR isolation. At present, the most important role in distinguishing the various nAChR subtypes of muscle-type, neuronal and non-neuronal (for example, on the immune cells), is played by α-conotoxins, neurotoxic peptides from the *Conus* marine snails. Contrary to nicotine and α-bungarotoxin, these peptides are of marine origin and at present play the most important role due to the high selectivity of naturally-occurring peptides and of their synthetic analogues in respect to distinct subtypes of neuronal nAChRs. A crucial role in the earlier characterization of the muscle-type receptor was played by protein α-neurotoxins from sea snakes (e.g., [1]); however, later, an overwhelming application was found for α-bungarotoxin, a long-chain α-neurotoxin from the many-banded krait *Bungarus multicinctus*, and especially its radioactive and fluorescent derivatives (see [2,3,4]). However, we will briefly consider new data on sea snake proteins homologous to α-bungarotoxin. There are several excellent recent reviews [5,6,7] relevant to the title of our article. We will mostly focus on the studies published in the last 10 years, including our work done in collaboration with various laboratories, employing different approaches for elucidating the ligand-receptor interactions.

The first cryo-EM structure of the muscle-type nAChR *Torpedo marmorata* was published by N. Unwin in 2005 [8], while high resolution structures of α-bungarotoxin complexes with this receptor and with neuronal α7 nAChR appeared only recently [9,10]. The previous X-ray structures of diverse compounds acting on the nAChR were solved for their complexes with the acetylcholine binding protein (AChBP), an excellent model of the ligand-binding extracellular domains of not only all types of nAChRs, but of all Cys-loop ligand-binding receptors (including the GABA_A_ receptors, glycine, 5HT3-serotonin and some other receptors). The first structure of the α-conotoxin complex with AChBP was established over 15 years ago [11]. A recent X-ray analysis of α-conotoxin LvIA analogue bound to AChBP was combined with computer modeling, alanine scanning of LvIA, analysis of binding of these analogs to α3β2 and α3β4 nAChRs, and mutations of the β2 subunit; as a result, some residues of this subunit responsible for the selectivity of α-conotoxin LvIA to α3β2 nAChR were identified [12].

Computer modeling of binding to AChBP is widely used to assess possible interactions of diverse low-molecular compounds, including those of marine origin, with various nAChR subtypes; then, for the most promising of them, such an activity is checked by experimental analysis of the interactions with AChBP, *Torpedo californica* and human α7 nAChR (see, for example, [13]). Thus, in this review we will try to show what a combination of different approaches (proteomic and transcriptomic studies of venoms, synthesis of naturally-occurring low molecular weight compounds and their analogues, synthesis of conotoxins, computer modeling, X-ray and electron microscopy analyses of the respective complexes with nAChRs or their models) reveals about the nAChR subtype selectivity and potential novel approaches for new drugs.

## 2. Marine Low Molecular Weight Compounds Targeting nAChRs

### 2.1. Biosafety Threat of Shellfish Poisoning with Nicotinergic Ligands

Harmful algae blooms are the source of various toxins produced mainly by dinoflagellates. The most dangerous human poisonings can occur as a result of toxin accumulation in marine organisms (mainly shellfish) directly consumed by humans. To get the whole picture on shellfish poisonings and similar events, please address the recent reviews by Anderson et al. [14] and Zingone et al. [15]. Marine algal toxins produce several toxic effects, including diarrhetic and various neurotoxic effects, with some effects being potentially attributed to nAChR antagonism by cyclic imine toxins from dinoflagellates: gymnodimines, spirolides, pinnatoxins, and some others (see Review by Molgo et al. [16]).

Biosafety-oriented studies in recent years demonstrated a variation of methods for the detection of the cyclic imines. The methods described in these papers include colorimetric [17] or chemiluminescent [18] assays based on competition with biotinylated α-bungarotoxin binding to *Torpedo* muscle-type nAChR immobilized on the microtiter plates. A similar technique based on flow-cytometry fluorimetric measurements of the *Torpedo* muscle-type nAChR, and *Lymnaea stagnalis* AChBP, immobilized on carboxylated microspheres, have also been suggested [19].

Macrocyclic imines (Figure 1) act potently as nAChR antagonists with picomolar affinity blocking muscle and some neuronal subtypes of the receptor in vitro (see Table 1) [20,21]. These molecules were thought to act at the orthosteric site of the receptor, since they prevented acetylcholine and other agonists binding and nAChR activation [22]. More recent research revealed that gymnodimine A and 13-desmethyl spirolide C (Figure 1) activate calcium flow through nAChRs [23]. Structures of macrocyclic imines are very complex and the chemical synthesis of such molecules is a challenging task hindering further research of spiroimine pharmacophores. However, Duroure et al. published their synthetic approach ten years ago [24].

Surugatoxins (Figure 1) were first found in the mollusks *Babylonia japonica* [25] and were shown to act at ganglionic nAChRs (Table 1) [26]. However, their exact affinities toward individual nAChR subtypes remain to be described in studies involving nAChRs assembled from clones of distinct subunits. Synthesis of a surugatoxin aglicone form has been described relatively recently [27], creating an opportunity to further toxin pharmacophore investigation.

An alkaloid from an ascidian species called lepadin B (Figure 1, Table 1) is an example of a nAChR ligand of marine origin which blocks, in submicromolar concentrations, neuronal hetero- and homopentameric receptor subtypes [28]. Total synthesis utilizing a new strategy of several lepadins, including lepadin B, was described in a recent paper [29].

### 2.2. Cembranoids from Marine Coelentera

Corals and marine jellyfish serve as a source of bioactive diterpenoids called cembranoids (Figure 1), which act on the nicotinic and some other Cys-loop receptors [33]. Cembranoid research has recently gained attraction mainly as a possible anticancer treatment [34]. It is now accepted that cembranoids can bind to nAChRs as ligands with a broad spectrum of activities: closed channel blockers, positive modulators and open-channel negative modulators (Table 1); the exact mode of their action is regulated by the concentration of the ligand [35]. On GABA_A_ receptors, which share some ligand-binding properties interactions with nAChRs [36], cembranoid eupalmerin shows positive allosteric modulation at the neurosteroid binding site [37]. Complex eupalmerin action on nAChRs could explain cell cycle arrest and apoptosis via the mitochondrial pathway in human glioma cell lines U87-MG and U373-MG [34], because it was previously shown that mitochondrial outer membrane possesses specific calcium-permeant nAChR subtypes [38]. Alternatively, c-Jun N-terminal kinase increased phosphorylation can arise from the nAChR modulation of the cell membrane through kinase cascades [39]. Moreover, the same mechanisms involving kinase cascades provide neuroprotection in neuroinflammatory conditions via interaction with α7 nAChR [39].

### 2.3. Nereistoxin from Marine Worm and Its Derivatives Used as Insecticides

Although nAChR ligands from marine sources could comprise a severe threat to humans, some of them are extensively used in agriculture as pesticides. Among others, nereistoxin (Figure 1) from marine annelids acts as a non-competitive nAChR antagonist at hybrid receptors consisting of insect and vertebrate nAChR subunits (Table 1) [32]. Its derivatives cartap (Figure 1), thiocyclam, thiasultap and bensultap act either as pro-drugs yielding nAChR-antagonising molecules or via direct action as nAChR non-competitive antagonists at the phencyclidine binding site [40] (for further information see the review [41]). Cartap has been shown to influence embryonic development, disrupting mainly skeletal organs development [42]. Case studies of human poisoning show that direct effects of nereistoxin on humans include rhabdomyolysis [43] and kidney failure [44]. Nereistoxin could reside in treated plant material such as tea leaves and pose a potential threat to consumer health [45]. Recent years have seen a rise of articles dedicated to methods of nereistoxin (along with cartap and other derivatives) detection and de-activation. Antidotes against nereistoxin derivatives poisoning include classical British anti-lewisite and sodium dimercaptopropane sulfonate [43]. Moreover, new chelators called “nanoreceptors” have been proposed recently to oppose the toxicity of nereistoxin [46].

Proposed methods for detection of nereistoxin and its derivatives include gas chromatography with electron-capture detector [47] and chromatomass-spectrometry [45,48,49]. Several methods of nereistoxin detection utilize the notable structural peculiarity of its molecule disulfide bond and its ability to react with gold. Specifically, quantum dots/gold nanoparticles fluorescent assay [50,51], electrochemical sensing based on transfer promotion effect on gold electrodes [52,53], and gold nanoparticles colorimetric assay [54] or DNA-switches [55] are selective nereistoxin (and its analogs) techniques, enabling fast and sensitive detection in the environment. It is interesting to note that varacin, a marine phenylethylamine derivative with an unusual sulfur-containing ring (Figure 1), has shown nAChR antagonistic properties at medium micromolar concentrations (Table 1) in one of our previous works [13]. Varacin was far more active against *Lymnaea stagnalis* AChBP, showing Ki 790 nM, suggesting that it or its analogs might be useful as pesticides.

### 2.4. Anabaseine, Its Derivatives and Other Marine Alkaloids with Clinical Perspective

Anabaseine (3,4,5,6-tetrahydro-2,3′-bipyridine) (see Figure 1) is the first isolated and identified toxic alkaloid produced by the carnivorous marine worms Nemertines [56]. These worms possess a variety of alkaloid, peptide or proteinaceous toxins that serve as chemical defenses against potential predators. Anabaseine is chemically similar to the tobacco alkaloid, anabasine, but possesses an imine double bond in the otherwise saturated piperidine ring and is classified as a non-selective nicotinic agonist, since it stimulates a broad spectrum of nAChRs [57]. Among three major forms of anabaseine (cyclic iminium, cyclic imine and the monocationic open-chain ammonium-ketone) co-existing in almost equal concentrations at physiological pH, the cyclic iminium is the form that avidly binds and activates vertebrate nAChRs [58].

The majority of substituted anabaseines display selective agonistic effects on the α7 nAChRs [57]. The 3-arylidene-anabaseines are of particular potential therapeutic interest because they have been shown to possess neuroprotective, as well as cognition enhancing, properties. Among them, 3-(2,4-dimethoxybenzylidene)-anabaseine (DMXBA) is the best studied, whose pharmaceutical code name GTS-21 (see Figure 1) refers to its origination as the 21st compound generated in a joint project by Gainesville (University of Florida) and Tokushima (Taiho Pharmaceuticals) scientists. GTS-21 is a lipophilic compound that readily passes across the gastrointestinal wall and the blood–brain barrier, and reaches peak concentrations in the blood and brain within a very short time [57]. GTS-21 was the first nicotinic agonist reported to stimulate α7 AChRs selectively (partial agonist), but it weakly antagonized α4β2 and other heteromeric nAChRs [59]. Tthe crystal structure of *Aplysia californica* AChBP bound with GTS-21 showed that the protein loop C is in the intermediate position, i.e., not utterly close to the orthosteric ligand-binding site, which is seen for full agonists [60]. Such conformational changes in the receptor structure may explain the mixed pharmacological profile of GTS-21 activity towards nAChRs.

GTS-21 was the first α7 agonist to enter clinical tests for possible treatment of cognition problems in schizophrenia, Parkinsonism and Alzheimer’s disease (AD) [57,59,61,62,63,64,65,66,67]. Clinical trials (phase II) of GTS-21 have been completed for such pathologies as schizophrenia [63], AD [67], tobacco use disorder and attention-deficit hyperactivity disorder (ClinicalTrials.gov identifiers: NCT00100165, NCT00414622, NCT02432066 and NCT00419445, respectively), but GTS-21 has not entered phase III clinical trials yet [65]. Future efforts may be directed to find out the most effective medicine formulation for GTS-21 and to study its pharmacokinetic limitations for effective improvement of cognition and for treatment of affective symptoms in schizophrenia [68,69].

In recent years, it has been shown that besides affecting cognitive functions, GTS 21 (and its active metabolite) effectively suppresses inflammation, demonstrating positive effects in a number of pathological conditions, such as acute ischemic stroke [70], sepsis [71,72], burns [73], acute lung injury [74], chronic obstructive pulmonary disease [75], airway inflammation [76] and atherosclerosis [77]. In clinical trials, the anti-inflammatory effects of GTS-21 were investigated in healthy human volunteers during experimental endotoxemia [78]. Although higher GTS-21 plasma concentrations significantly correlated with lower cytokine levels, the highest dose tested to be safe in humans did not result in significant differences in inflammatory mediators between the GTS-21- and placebo-treated groups.

As the basic mechanism of the observed positive pharmacological properties of GTS-21, most researchers describe the activation of the cholinergic anti-inflammatory pathway (and similar mechanisms) through α7 nAChR stimulation in immune cells [71,73,74,75,77,78,79]. The molecular mechanisms by which GTS-21 exerts its anti-inflammatory actions in immune cells have been attributed to the blocking effects of the important pro-inflammatory transcription factor NF-kB, nuclear factor kappa-light chain-enhancer of B cells [80], activation of the JAK/STAT pathway [81], or increased levels of intracellular cAMP [74]. However, many details of α7 nAChR involvement in GTS-21 effects on inflammatory pathways remain unidentified, such as the role of the receptor conductive/non-conductive states [82], some anti-inflammatory effects of this partial agonist independent of α7 nAChRs [83], and induction of pro-inflammatory IL-6 release by non-immune cells [84].

Besides pro-cognitive and anti-inflammatory effects, GTS-21 also displays neuroprotective [85], antinociceptive [86], antiproliferative [87], immunomodulatory [88] and metabolic homeostasis-regulatory properties [79]. The advantage of GTS-21 treatment was reported for a wide spectrum of pathological conditions, providing a basis for future development of α7 nAChR-gated therapy with anabaseine derivatives.

A structurally-different set of anabaseine derivatives was constructed; its 3-pyridinyl moiety was replaced with a 3-mono- or 3,5-disubstituted benzene ring as a means to gain selectivity for the α3β4 nAChR subtype [89]. The synthesized compounds activated the α3β4 nAChR with high affinity, while eliciting a negligible response at the α7 subtype and no effect at the α4β2 subtype. 5′-Phenylanabaseine, acting as a conditionally silent agonist, comprises another class of anabaseine derivatives worth mentioning [90], as they might act as an alternative pathway to the α7 nAChR-targeting drug-design.

It is also worth mentioning that some other marine alkaloids acting on nAChRs are now introduced not only as a valuable tool for fundamental research, but also as potential drugs. In our work, we have demonstrated that makaluvamine G from the marine sponge *Zyzzia fuliginosa* (Figure 1) binds both to orthosteric and allosteric sites of the muscle-type nAChR (Table 1) [91], thus being a perspective molecule for the design of allosteric drugs targeting these receptors, especially for treating slow-channel congenital myasthenic syndromes [92]. 

As shown above, nAChR agonists from marine sources could have a tremendous effect on the drug design field. A few novel marine nAChR agonists have been discovered in recent years, including 6-bromohypaphorine from the nudibranch mollusk *Hemissena crassicornis,* targeting α7 nAChR [93] and molleamines [94]. The latter have recently been shown to be partial agonists at α3- and α6-containing nAChRs expressed in dorsal root ganglions [94]. Moreover, signs of antinociceptive activity and the lack of acute toxicity make molleamines promising hits for drug design.

## 3. Marine Origin Peptides Targeting nAChRs

### 3.1. A Start of the “Conotoxin Era”

Since the discovery of nAChRs, for a long time they have been studied exclusively with the use of low-molecular compounds (nicotine, carbamoylcholine, etc.) and protein α-neurotoxins from snake venoms (α-bungarotoxin, α-cobratoxin, etc.), while the latter played a decisive role in the isolation of purified preparations of muscle-type nAChR and the identification of the first neuronal subtype of the receptor in the brain. However, in 1981, the first three peptide blockers of neuromuscular transmission, called conotoxins GI, GIA, and GII [95], were isolated from the venom of the marine predatory mollusk *Conus geographus*, which acted through the muscle-type nAChR [96]. This event marked the beginning of a whole “conotoxin” era in the study of nicotinic receptors.

In general, the venoms of *Conus* mollusks turned out to be an invaluable source of biologically active compounds, mainly of a peptide nature, named conopeptides, which act on a wide variety of targets (see the functional activity of various conopeptides in early reviews, for example, [97]). They are classified in the gene superfamilies (A, B, C, D, etc.) by the similarity between the signal sequences of the precursors. Conopeptides rich with disulfides (from two) conotoxins are also subdivided into structural families by the pattern of cysteine residues in the mature peptide (I, II, III, IV, etc.). Finally, an additional classification in the pharmacological families occurs by the molecular targets and physiological activity (α, γ, δ, ε, etc.) (for a more complete classification, see the early reviews, for example [7,98]. 

The conopeptides from several such families act on various subtypes of nAChR, the most famous and studied of which are the α-conotoxins having cysteine framework I with two disulfide bonds (C^1^–C^3^ and C^2^–C^4^) (Figure 2). During the followinng 25 years after the discovery of the first α-conotoxins, nAChR-targeting peptides with the different cysteine patterns were identified in the venoms of various Conus species—αA-, αS-, αD-, αL-, αS-, and ψ-conotoxins (Figure 2 and Figure 3). The last decade has supplemented this list with two new families: αB [99] and αO-conotoxins [100] (see Figure 2); however, these families are represented by one or only a few peptides, which, as a rule, have not received further scientific development. As an exception, we can mention the discovery of a new αS-conotoxin, GVIIIB, which was a highly selective inhibitor of α9α10 nAChR [101], αM-conotoxin MIIIJ, which, like ψ-conotoxins, belongs to the M-superfamily and also blocks muscle nAChR [102], αO-conotoxin GeXIVA [100], with good prospects for medical use (see below) and several αD-conotoxins. Of the two isolated dimeric αD-conotoxins, GeXXA [103] (Figure 3) and PiXXA [104], the first was shown to be a non-selective blocker of both neuronal and muscle receptor subtypes, while for the second, inhibitory activity was revealed only towards human α7 nAChR (IC_50_ 6.2 µM). Another αD-conotoxin Lt28.1, with a cysteine pattern (-C-C-C-CC-C-C-C-C-C-) unusual for this family, was cloned from *Conus litteratus*, expressed in a yeast system, and showed low affinity (IC_50_ 3.0 µM) only towards α9α10 nAChR [105].

It should be noted that, in the last decade, genomic and transcriptomic approaches have led to the identification of a large number of new potential peptides from the mentioned families in the venoms of dozens *Conus* species, but not all of them have been synthesized and studied for their biological activity and therefore remain outside the scope of this review. Genome-derived peptides also leave open the question of post-translational modifications and especially of “natural” folding due to the formation of disulfide bonds. For this reason, in structural and functional studies of such compounds, all possible isomers are often synthesized. In particular, for α- and αO-conotoxins, a large number of studies provide results obtained using not only the “classical” globular form of conotoxins (C^1^–C^3^ and C^2^–C^4^), but also the ribbon (C^1^–C^4^ and C^2^–C^3^) and beads (C^1^–C^2^ and C^3^–C^4^) ones. However, these works, in general, also remained outside the scope of this review.

Isolated from *Conus* venoms or revealed by genetic methods and then synthesized, several dozen α-conotoxins are additionally subdivided into groups according to the number of amino acid residues in their first and second loops (between Cys2-Cys3 and Cys3-Cys4, respectively). Five such groups are traditionally known (3/4, 3/5, 4/4, 4/6, 4/7), and spatial structures of a large number of their representatives were established by NMR or X-ray analysis (Figure 3). The first representative of the new 5/5-group, α-conotoxin AusIA, was isolated from the venom of *C. australis*, but its globular and ribbon forms showed low affinity (IC_50_ ≥ 10 μM) only to α7 nAChR [107]. In addition, a new conotoxin Pl168 with a cysteine pattern characteristic of α-conotoxins was discovered recently, but with unusual 4/8-spacing, for which, however, no nAChR subtype has been identified as a target [108].

It should be noted that, over the past decade, the blocking of distinct nAChRs as a second target has been revealed for conopeptides belonging to other families and superfamilies. Here, we can mention the detected interaction of the μ-conopeptide CnIIIC (Figure 3), a powerful blocker of several types of potential-dependent sodium channels, also with α3β2, α7 and α4β2 nAChR subtypes [109]. Another example is the inhibition of muscle and α7 nAChRs by two conorfamides As1a and As2a, in addition to their expected interaction with distinct subtypes of ASICs [110]. Besides, the conotoxin TxVC (KPCCSIHDNSCCGL*), isolated from the *C. textile* venom and related by its structural parameters to the T-superfamily of conopeptides, did not show a noticeable interaction with sodium, potassium and calcium channels, but inhibited α4β2 and α3β2 nAChRs with IC_50_ 343.4 and 1047.2 nM, respectively [111].

The rapid development of the “conotoxin” direction over the past four decades has pushed the search for new peptide tools for studying nAChRs from other marine sources into the background. Up to the present day, no effective cholinergic ligand of a peptide nature has been isolated from marine sources other than *Conus* venoms.

### 3.2. Discovery of New Conotoxins Targeting nAChRs

Despite the large number of α-conotoxins that have already been discovered and studied (see, for examples, early reviews [6,112,113]), the search and characterization of new representatives of this conopeptide family is currently actively continuing, both by methods of direct isolation from venoms and detection in the genome with subsequent peptide synthesis. This partly results from the fact that this family of conopeptides is represented in each *Conus* species venom and replenishment is due to the study of new species of mollusks, as well as to the identification of minor α-conotoxins in previously studied ones. For the latter, the application of genomic approaches plays a crucial role.

One of the main aspects of the relevance of the search and characterization of new α-conotoxins remains the diversity of their action on different nAChR subtypes compared to protein ligands and a noticeably higher selectivity to them as compared to low-molecular weight compounds. This still makes α-conotoxins an exceptional tool for fundamental studies of different nicotinic receptor subtypes, and also opens up new prospects for practical use either as appropriate markers or drugs selectively blocking a specific target involved in a respective disease.

Over the past decade, a number of new α-conotoxins have been identified, which are summarized in Table 2. Most of them are representatives of the 4/7-group of α-conotoxins. Alpha-peptide, RegII and LsIA isolated from *C. tinianus*, *C. regius* and *C. limpusi* venoms, respectively, showed non-selective interaction with several neuronal nAChR subtypes [114,115,116]. The same ability was revealed for synthetic Mr1.7 [117], derived from the *C. marmoreus* venom duct cDNA library [118] (Table 2). Highly selective among the discovered α4/7-conotoxins were TxIB towards the α6/α3β2β3 receptor subtype [119], LvIA and Lt1.3 towards the α3β2 nAChR [120,121], and LvIF and Bt1.8 with high nanomolar affinity for α3β2 and α6/α3β2β3 subtypes [122,123], while α-conotoxins Lo1a and BnIA acted on the α7 nAChR (although with micromolar affinity) [124,125] (Table 2). The latter, isolated from the *C. bandanus* venom, has the same sequence as α-conotoxin Mr1.1, previously identified from the cDNA library of *C. marmoreus* [126]. Interestingly, the ribbon isomer of α-conotoxin Lt1.3 loses its high affinity for α3β2 nAChR, but gains inhibitory activity towards the GABA_B_ receptor [121]. For most of the mentioned α-conotoxins (RegII, LsIA, Mr1.7, LvIA, Lt1.3, Bt1.8), the key amino acid residues were revealed that determine their selectivity to the respective nAChR targets (Table 2).

Surprisingly, among the newly discovered α4/7-conotoxins, one compound exhibited agonistic properties towards nicotinic acetylcholine receptors. α-Conotoxin MrIC (Table 2) became the first complete agonist of a peptide nature of human α7 nAChR in SH-SY5Y cells in the presence of this receptor subtype silent agonist PNU120596 [127,128].

Several novel α-conotoxins from other groups have also been discovered and investigated in recent years. Thus, in the venom of *C. ermineus*, two new 4/4-conotoxins, EIIA and EIIB, containing an increased number of post-translational modifications, were identified by mass spectrometry, and were shown to be highly selective blockers of the muscle-type nAChR from *Torpedo* [132,133]. Noteworthy, EIIA conotoxin distinguished two binding sites on this receptor (Table 2). An interesting role of post-translational modifications in α-conotoxins was revealed with the new α4/7-conotoxin, TxIC [148]. The natural toxin (ROQCCSHOACNVDHPγIC*) found in the venom of *C. textile* showed an almost complete absence of inhibitory activity towards different nAChR subtypes (IC_50_ > 50 μM), and its analogue, without appropriate modifications (RPQCCSHPACNVDHPEIC*), micromolar affinity (IC_50_ 2.1–5.4 μM) towards neuronal α3ß4, α3ß2, and α3α5β2 receptors. Another two novel α4/4-peptides were isolated from the venom of *C. purpurascens*: α-conotoxin PIC and its [P7O]-modified form. They blocked rat muscle receptors and human neuronal α3β2 nAChR with moderate affinity (IC_50_~1 µM) [138] (Table 2).

A few novel conotoxins from the 4/6-group have also been characterized recently. Thus, α-conotoxin ViIA, identified in the cDNA library of the *C. virgo* venom duct, showed moderate affinity (IC_50_ 845.5 nM) only to α3β2 nAChR [137]. Another α-conotoxin, TxID, from the same group (see Figure 3), revealed high selectivity to α3β4 receptor (IC_50_ 12.5 nM), a rare selectivity as compared to other α-conotoxins [135]. For both conotoxins, the key amino acid residues determining their affinity toward respective nAChR subtypes were identified (Table 2). α-Conotoxin VnIB was also found in the genomic DNA, synthesized, and showed a highly selective affinity for α6β4-containing nAChRs (IC_50_ 5.3–12 nM depending on the human/rat species selectivity [140] (see Table 2).

α-Conotoxin MilIA, a new representative of the 3/5 group, comprising most of the peptides that inhibit muscle nAChRs, was recently isolated from the venom of *C. milneedwardsi* [141]. Surprisingly, the natural toxin showed very low affinity for both subtypes of the muscle receptor (IC_50_ 11–13 μM), unlike some of its synthetic analogues with mutation at 9, 10 and 11 positions (Table 2). Another “classic” muscle nAChR-acting α-conotoxin, CIA, was identified in the transcriptome of the *C. catus* venom duct, and then synthesized together with α4/7-conotoxins CIB [139] and CIC [143]. In addition to the nanomolar affinity towards the muscle receptor subtype, CIA unexpectedly showed a micromolar affinity for α3β2 nAChR, while the main target for CIB was the α3β2 nAChR receptor [139] (Table 2). The structural feature of the α4/7-conotoxin CIC was the presence of an elongated N-terminus, but it did not affect the main activity of this peptide towards α3β2 and α6/α3β2β3 nAChRs [143].

A number of new peptides, α3/5-conotoxin GIB, similar in structure to the first discovered conotoxins GI and GIA [95], classic α4/7-conotoxin G1.5, and a peptide G1.9 (ECCKDPSCWVKVKDFQCPGASPPN) with an unusual cysteine 4/8-spacing [142], were identified in the venom of *C. geographus* by proteomic and transcriptomic analyses. Their synthetic analogues (in globular and ribbon isoforms) showed the expected affinity of GIB towards the human muscle α1β1εδ receptor, and for G1.5 towards a number of neuronal receptor subtypes with the best affinity for α3β2 (IC_50_ 35.7 nM) (Table 2). The unusual 4/8-peptide G1.9, as well as the previously discovered similar conopeptide Pl168 [108], did not show the ability to interact with any of the tested nAChR subtypes.

The studies on the expression of venom mRNA in embryos of *C. victoriae*, which led to the identification of two new sequences, Vc1.2 and Vc1.3 [129], which are close to the structure of the well-studied α-conotoxin Vc1.1, were unexpected. Interestingly, Vc1.2 has a different selectivity profile, having lost the affinity towards the α9α10 nAChR subtype (Table 2), while for Vc1.3 there was no noticeable ability to interact with neuronal nAChR subtypes. The new selectivity of Vc1.2 is determined precisely by those amino acid residues in the first and second loops of the molecule that are different from Vc1.1 (Table 2).

The history of the first conotoxin from the L-superfamily acting on nAChR αL-conotoxin lt14a, initially identified in the cDNA library of the *C. litteratus* venomous duct in 2006 [144], is unusual. For its synthetic form, an analgesic effect was detected in the “hot plate” test in mice, and inhibition of neuronal nAChRs in PC12 cells was shown. Since that time, several studies have been carried out on the analgesic capabilities of this peptide and its NMR structure has been obtained [145] (see Figure 3), but the distinct neuronal nAChR subtypes on which it acts and the affinity for them have not yet been identified (Table 2). A little later, DNA cloning identified another conopeptide with the same cysteine pattern Pu14a, which blocked some subtypes of nAChR with low affinity [146] (Table 2), and for which the NMR structure was also obtained [149].

Special mention should be made of the discovery, synthesis and structure-functional studies of the first representative of the O1-superfamily of conopeptides, effectively interacting with the nAChR, namely, αO-conotoxin GeXIVA from *C. generalis* [100]. It interacted with α9α10 nAChR with nanomolar affinity, while the affinity for the other receptor subtypes was at least two orders of magnitude lower. Interestingly, this selectivity and high affinity were shown by all three synthesized possible isomers of GeXIVA (IC_50_ 22.7, 7.0, and 4.6 nM for globular, ribbon and beads, respectively) (Table 2); the ribbon and beads isoforms showed higher affinity to the target, as well as much sharper 1D-NMR spectra. The study of the interaction mechanism of GeXIVA has shown its possible binding to the allosteric site on α9α10 nAChR, although in micromolar concentrations this peptide can also interact with the receptor orthosteric sites, displacing α-bungarotoxin at the α9 subunit extracellular domain, as well as at the *A. californica* AChBP [150].

Another representative of the O-superfamily, namely GeXXVIIA, was isolated from the venom of the same mollusk [147]. This naturally occurring compound turned out to be a disulfide-bound homodimer. Refolding of the synthesized linear peptide resulted in four different monomeric isomers of this conotoxin, which inhibited various neuronal and muscle nAChRs, with the highest affinity towards α9α10 and α1β1εδ subtypes. The linear form of the monomer also demonstrated a high inhibitory activity against the same two subtypes (IC_50_ 16.2 and 774 nM, respectively). Interestingly, the affinity to the first subtype is determined mainly by the N-terminal part of the molecule, and to the second one by the C-terminal part (Table 2).

### 3.3. Studies on Molecular Bases of nAChR Subtypes Selectivity with Conotoxins

The new conotoxins listed in the Table 2, as well as previously discovered ones, have been often used in the past decade in fundamental studies on the nAChRs. Those studies have concerned a variety of aspects of the functioning of nicotinic receptors. Here we will focus on one such direction, the elucidation of the molecular basis of selectivity of different nAChR subtypes with the aid of conopeptides. One of the approaches was to obtain a series of analogues of a selected conotoxin and of mutant forms of the respective receptor and to evaluate the changes in the affinity of their interaction. It makes it possible to establish point contacts between the amino acid residues of the ligand and the target, and build computer models of the receptor–ligand complexes.

In particular, in continuation of the work [151], which revealed key residues in α-conotoxin RgIA responsible for its affinity to α9α10 and α7 nAChR subtypes (Asp5, Pro6, Arg7, Arg9), a large series of analogues with substitutions for various residues at positions 7, 9–11, 13 was obtained [152]. As a result, the fundamental role of Arg7 was confirmed. In addition, the importance of the chain length, but not of the charge in position 9, was revealed. Finally, a set of analogs mutated at positions 10, 11 and 13, with increased affinity for both the human and rat receptors, was obtained.

“Reversed” mutagenesis from the receptor side first revealed an amino acid residue in the α9 subunit (T56) responsible for a high affinity for the rat α9α10 receptor, more than 300 times higher than the affinity for the human one [153]; then, it was convincingly proved that the interaction of α-conotoxin RgIA occurs at the α10(+)/α9(−) interface, rather than at α9(+)/α10(−) [154]. A similar conclusion was made also for α-conotoxin Vc1.1, based on computer modeling, and tested on a mutant receptor and two designed and synthesized analogues of conotoxin [155]. In addition, the use of computer modeling, supported by two cycles of minimal side chain replacement in conotoxin Vc1.1 [156], led to the assumption of the probable interaction of S4, Y10 and D11 conotoxin residues with D166/D169, N107 and R81/N154 of α9 subunit, respectively, as well as to the design of two analogues of Vc1.1, [S4Dab, N9A]Vc1.1 and [S4Dab, N9W]Vc1.1, with an affinity for α9α10 nAChR more than 20 times higher than that of a natural peptide. In general, the use of various computer approaches has recently become more common and allows the design of new analogs of conotoxins with specified properties. An example is the application of the ToxDock docking algorithm for the successful design of α-conotoxin GID analogs with reduced affinity for its second target [157]. Another example is the use of free perturbation energy calculations to perform an amino acid mutation scanning of α-conotoxin LvIA and predict mutations resulting in the increase of its selectivity towards α3β2 nAChR [158].

Such mutagenesis studies, usually supported by computer modeling, have revealed key amino acid residues on the part of the conopeptides, and in some cases also on the part of the targets, responsible, in particular, for differences in the interaction of α-conotoxin PeIA with α6/α3β2β3 and α3β2 nAChRs [159]. Mutagenesis of the receptor α6 subunit made it possible to identify amino acid residues responsible for the selectivity of α-conotoxin BuIA towards α6/α4β2β3 nAChR [160]. Similarly, mutagenesis of the α3 and β2 subunits, combined with computer modeling, allowed a mechanism of antagonistic action of this conotoxin on the human α3β2 receptor to be proposed [161]. Alanine-scanning was performed for α-conotoxin AuIB, which revealed the residues determining its selective affinity for the α3β4 nAChR, and the computer simulation suggested their partners in the β4 subunit, which was confirmed by site-directed mutagenesis [162]. Mutagenesis in the β2 subunit revealed three residues (T59, V111, F119) that maximally affect the affinity (both increase and decrease) of α-conotoxin LvIA towards the α3β2 receptor [163]. Alanine-scanning mutagenesis has also been carried out recently for some “muscle” α-conotoxins, which identified the residues determining their selectivity to the α1β1εδ nAChR [164,165]. Similarly, the residues important for selective targeting of the α3β4 and α3β2 receptors were identified as a result of alanine-scanning mutagenesis of α-conotoxin RegIIA [130]. 

Similar studies have also made it possible to prepare potent and selective analogues, for example, for the α6/α3β2β3 nAChR, based on α-conotoxin PeIA (IC_50_ 0.223 nM) [166] or for human α3β2 receptor based on α-conotoxin RegIIA (IC_50_ 13.7–27.1 nM) [167]. Another example is an [S9A]TxID analog which effectively distinguished the α3β4 from α6/α3β4 nAChRs (IC_50_ 3.9 vs. 178.1 nM) [134]. Computer analysis became the basis for the synthesis of a large series of analogues of α-conotoxin PeIA; as a result, the species (rat/human) specific ligand-binding motifs in relation to α6β4 receptors were revealed, and analogues with significantly higher affinity for the rat nAChR were purposefully designed [168].

In general, elucidation of the molecular basis of the species selectivity of some α-conotoxins towards the distinct nAChR subtypes (mainly in a human/rat pair) has become one of the most important areas of recent research. Thus, the use of α-conotoxin RegIIA and its analog [N11A,N12A]RegIIA revealed that the reason for the almost 70-fold difference in the affinities to human and rat α3-containing receptors is a single amino acid residue in the 198 position of the α3 subunit (Glu—human, Pro—rat) [169]. Another analogue of this peptide, [H5D]RegIIA, as well as [K11A]TxIB, demonstrated two orders of magnitude higher affinity for rat α7 nAChR than for the human one, which was caused by two residues at the positions 183 and 185 of this receptor subunit [170]. Interestingly, the same set of amino acid residues of the α7 subunit supplemented with residue 140 (sequence alignment from [170]) was found to be responsible for the same specie selectivity for another designed conopeptide [Q1G,ΔR14]LvIB [171].

Crystallographic studies of the water-soluble spatial homologues of ligand-binding receptor domains AChBPs and recombinant extracellular domains of individual subunits and of their complexes with various ligands, including conopeptides, provided significant progress in the structural studies of nAChRs and, first of all, of the fine organization of their ligand-binding sites. Starting with the first structure of the complex of AChBP from *A. californica* with an analog of α-conotoxin PnIA [11], more than a dozen similar crystal structures have been obtained to date (some of them with α-conotoxins from different groups are represented in Figure 4). Over the past decade, among other things, the structures of *A. californica* AChBP with α-conotoxin GIC have been published, which made it possible to identify determinants responsible for its exclusive selectivity to α3β2 nAChR [172]. The structure of AChBP with α-conotoxin LvIA revealed the mechanism of its selectivity towards different nAChR subtypes [173]. Similarly, X-ray structure of the same protein with α-conotoxin PeIA showed molecular determinants for its species selectivity (human/rat) towards α6β4 nAChR [174], and with analogues [N9A]LvIA and [D11A]LvIA, the first of which distinguished the rat α3β2 and α3β4 receptors with the affinities differing more than three orders of magnitude, made it possible to reveal the key amino acid residues in the β2-subunit responsible for this difference [12].

Several structures of conotoxin complexes with AChBP from *L. stagnalis* have also been obtained. In particular, the structure of the complex of this protein with α-conotoxin LsIA resulted in identification of the minimum pharmacophore regulating the α3β4 antagonism [131]. Crystallization with *L. stagnalis* AChBP of the two isoforms (globular and ribbon) of the unusual α5/5-conotoxin AusIA, equipotential to the α7 nAChR in combination with the substitutions in the peptide, made it possible to explain this pharmacological feature due to the presence of the fifth amino acid residue in the first loop of this conotoxin, and also to determine the key residues responsible for the α7 selectivity (Table 2) [136].

The crystal structure of the first marine origin peptide (α-conotoxin RgIA) in a complex with the extracellular domain of the nAChR subunit was recently established [175]. Despite the fact that it was only a monomer of the extracellular domain of the α9 subunit of the human receptor, computer simulations on full-size human α9α10 nAChR suggested a favorable binding of RgIA at α9/α9 or α10/α9, rather than at the α9/α10 interface, in accordance with previous mutational and functional data.

Interestingly, all the obtained structures of complexes with conopeptides were characterized by a similar arrangement of conotoxins in the orthosteric binding site (under the C loop) (Figure 4). However, each conotoxin revealed its own characteristic set of contacts (sometimes overlapping) with amino acid residues of the AChBPs or subunit domain (for a detailed comparison of these pair determinants see, for example, reviews [176,177]). It is worth noting that no structure has yet been obtained with the location of the peptide ligand outside the orthosteric binding site, although sufficient data has already been collected in the literature on the possible binding of some conotoxins on allosteric sites of nAChRs, for example, for α-conotoxin MrIC at the α7 receptor [178], or αO-conotoxin GeXIVA at the α9α10 nAChR [100]. 

In recent years, there has been a breakthrough in the use of cryo-electron microscopy for nAChRs, which allowed resolution of the structures of the muscle-type receptor from the *Torpedo* ray electric organ [8], the neuronal human α4β2 nAChR in complex with nicotine and antibody fragments specific for the β2-subunit [179], the α3β4 receptor also in complex with agonists but already in a lipid environment [180], as well as α7 nAChR in three different states (apo, agonist-bound and agonist/PNU-120596 bound) [181]. It is important that over the past two years, cryo-electron microscopy structures of nicotinic receptors have also been obtained in complex with the polypeptide antagonist α-bungarotoxin from the banded krait venom, namely the muscle-type receptor from the *Torpedo* electric organ [9] and the neuronal human α7 nAChR [10] (for more detailed high resolution structural studies of full-size nicotinic receptors, see, for example, the review [182]). Probably, the respective structures with peptide ligands should be expected in the near future.

### 3.4. Medical Perspectives for Conotpeptides Targeting nAChRs

The possibility of effective practical use of marine origin peptides was clearly demonstrated by the development of Prialt^®^ (Ziconotide), which is a synthetic analogue of ω-conotoxin MVIIA blocking N-type voltage-gated calcium channels, successfully used to relieve pain in cancer patients (see, for example, reviews [183,184,185]). However, we have to state that no drugs based on the conotoxins targeting nAChRs have yet been developed. Nevertheless, the basis for the search for such compounds with prospects for clinical use are numerous studies carried out in vitro and in vivo, in which the analgesic and anticancer properties of quite a large set of conotoxins were demonstrated. A detailed description of all conopeptides studied in this aspect would require a separate review (the list of conopeptides under different stages of preclinical trials can be found in the reviews [7,186,187]); therefore, here we will briefly mention only the main compounds that are currently considered as a possible basis for developing successful drugs.

The design of analgesics based on the nAChR-targeting conopeptides has the longest history and is primarily associated with α-conotoxins Vc1.1 and RgIA. Since their discovery, the analgesic effects of these peptides have been shown in a number of tests in vivo on rodents, which were initially related only to their highly effective inhibition of α9α10 nAChR. Soon, however, it was found that the affinity of both peptides to the same human receptor subtype is significantly lower [153,155], but this problem has been solved over the past decade by designing a number of analogues (including those containing non-canonical amino acid residues, see the special section of the review [188]) with nanomolar and even sub-nanomolar affinity towards the human α9α10 receptor [189,190,191].

However, a second target of α-conotoxins Vc1.1 and RgIA GABA_B_ receptors was identified (as well as for some other α-conotoxins [192]) and a different, more complex, mechanism of analgesic action was proposed through potent inhibition of N-type calcium channels via GABA_B_ receptor activation. The role of cholinergic and GABA_B_-ergic systems in the analgesic action of these conopeptides is still being discussed and is reflected in numerous experimental studies and reviews (see, for example, [184,193,194]). Perhaps the analgesic mechanism is even more complex, since the first data on the modulation of G protein-coupled inwardly rectifying potassium (GIRK) channels by these conotoxins via the activation of GABA_B_ receptors have appeared [195]. Meanwhile, over the past decade, new data have appeared on analgesic activity in various animal pain models for other conopeptides, in particular, for all three isomers of αO-conotoxin GeXIVA [100,196,197], αL-conotoxin lt14a [198,199], α-conotoxins AuIB, MII [200], BuIA and its analogues [201], and LvIA [202].

In the last decade, studies have also been published evaluating the possibility of using some conotoxins as anticancer agents. At the moment, such prospects are due only to experiments in vitro demonstrating the ability to suppress the growth of various cancer cells with a number of compounds: αO-conotoxin GeXIVA [203], α-conotoxins MII, PnIA, ArIB, RgIA [204], AuIB [205], and TxID [206]. This gives some hope for the development in the near future of effective drugs based on conopeptides for the treatment of cancer. 

## 4. Marine Protein Ligands of nAChR—Still an Open Field for Research

### 4.1. Three-Finger α-Neurotoxins

Protein ligands of marine origin targeting nAChRs are represented mostly by α-neurotoxins from the venoms of sea snakes. Sea snake toxins were among the first ones purified from snake venoms and sequenced [207]. Erabutoxin b from the venom of the sea snake *Laticauda semifasciata* was the first toxin shown to bind at the cholinergic receptor site [208], and the first three-finger toxin for which 3D structure was established by X-ray analysis [209,210]. Accordingly, these toxins played an important role in early research on the α-neurotoxins, but their role has diminished over time. The studies of sea snake neurotoxins are summarized in the review [211]. Recently, the 3D structures of *Laticauda semifasciata* erabutoxin a complexes with α7 and muscle type nAChR were studied by molecular modeling [212]. For this purpose, erabutoxin a was docked to a homology model of the α7 nAChR extracellular domain and to the αγ and αδ interfaces of the muscle type *Torpedo* nAChR ((α_1_)_2_βγδ) structure [9] using Rosetta protein–protein docking protocol [212]. It was found that the toxin interacted with the binding sites of α7 and muscle-type nAChR through D31, F32, and R33 amino acid residues. All these interactions were shown earlier to be important for erabutoxin a binding to the muscle-type nAChR [213].

All snake venom α-neurotoxins inhibiting nAChRs are divided into two structural types, so called short-chain and long-chain ones [214]. Short-chain α-neurotoxins consist of 60–62 amino acid residues with the spatial structure stabilized by four intramolecular disulfide bridges; they inhibit the muscle type nAChRs with high selectivity, and so far, are the most selective protein ligands of the muscle type nAChRs. Long-chain α-neurotoxins comprise 66–75 amino acid residues, with the scaffold stabilized by five disulfide bridges, and these toxins additionally block α7 nAChRs and α9α10 nAChRs. Despite this non-selectivity, long-chain α-bungarotoxin is often used as a marker of the muscle-type and α7 nAChR subtypes.

### 4.2. Proteomic and Transcriptomic Analyses of Sea Snake Venoms

The development of new analytical methods greatly changed the field of venom investigations during the last decade and the studies of sea snake venoms being carried out mainly by proteomic approach. Thus, identification of new toxins by this strategy is based on the previous knowledge of the structure and function of the related proteins; therefore, it is difficult to expect the discovery of completely new toxins using this method. About a dozen works employing proteomic and transcriptomic methods for characterization of the sea snake venoms were published during last decade. The venoms of the spine-bellied sea snake *Hydrophis curtus* and the annulated sea snake *Hydrophis cyanocinctus* aroused the greatest interest. Both venom proteomes [215,216,217,218,219,220] and venom gland transcriptomes [219,220,221] were characterized for these species of different geographical origin. The main finding of all these works is that the α-neurotoxins of both short-chain and long-chain types represent the most abundant toxin family in these snakes. The content of α-neurotoxins varied from 26 to 88% depending on the origin of snakes. The studies of other species including the pelagic yellow-bellied sea snake *Pelamis platura* [222], the olive sea snake *Aipysurus laevis* [223], the beaked sea snake *Hydrophis schistosus* [217], the yellow-lipped sea krait *Laticauda colubrina* [224], and the yellow-bellied sea snake *Hydrophis platurus* [215] showed similar results; α-neurotoxins were the main constituents of the venoms. 

The original approach was used to study venom proteins of *Lapemis hardwickii* (synonym to *Hydrophis curtus*) [225]. By using a cDNA T7 phage display library constructed from venom gland mRNA, several new toxins belonging to different families were found. Four new short-chain α-neurotoxins (SNT-1, 2, 3, 4) were identified, cloned and expressed in *Escherichia coli*. The analgesic activity of purified toxins was studied using the acetic acid-induced writhing test. All toxins manifested analgesic activity at a dose equal to 1/4 LD_50_, with SNT-4 being the least toxic to mice and showing the highest analgesic activity. The molecular modeling performed by the authors showed that the toxins bind the muscle type nAChR; however, no explanation has been proposed for the possible molecular mechanism of analgesic activity.

### 4.3. Non-Snake Proteins of Marine Origin Acting on nAChRs

Just a few proteins affecting nAChRs were identified in other marine organisms. Thus, a tentacle venom extract from a scyphozoan jellyfish *Aurelia aurita* produced neurotoxic effects in ghost crabs *Ocypode quadrata* [226]. After separation of extract by reversed-phase HPLC, several fractions were obtained; two of them showed neurotoxic effects and were analyzed further. One fraction, comprising mainly the protein with a molecular mass of about 90 kDa, in electrophysiological experiments potently inhibited ion currents induced by acetylcholine in the fetal and adult muscle type nAChRs. The nAChR inhibition was concentration-dependent and completely reversible, the IC_50_ values being different for fetal (1.77 μg/μL) and adult (2.28 μg/μL) receptors. This fraction showed no phospholipase A_2_ activity. 

It was shown earlier that venom of the lionfish *Pterois volitans* contains acetylcholine and a toxin that affects neuromuscular transmission [227]. In the recent study, the effects of *P. volitans* venom on the nAChRs and dopaminergic neurons were characterized [228]. It was found that the venom inhibited the response induced by acetylcholine in the human α3β2 nAChR by about 50%, but produced no effect on the response of the human α7 receptor. After the venom separation by reversed-phase HPLC, two main fractions were obtained. Only one less polar fraction inhibited the response of neuronal α3β2 nAChR by 57%, and this inhibition was reversible. Electrophoretic analysis showed that the active fraction produced three intense bands of ∼39.2, 15.7 and14 kDa, the 15.7 kDa band being more intense.

Peptide neurotoxins from marine snails of the *Conoidea* superfamily were described in the above section. The *Conoidea* superfamily comprises many genera and the overwhelming majority of neurotoxins were isolated from the snails from the family *Conidae*. the study of the *Turridae* family as a source for nAChR ligands was initiated only relatively recently [229,230]. Along this line, the venom duct extract from the East Pacific species *Polystira nobilis* was separated by reversed-phase HPLC and the biological activity of six selected fractions was analyzed by electrophysiological measurements on human α7 AChRs expressed in *Xenopus laevis* oocytes [231]. One of the studied fractions inhibited the response induced by acetylcholine by 62%. Further purification resulted in the peptide which, at 5.6 μM, strongly and irreversibly inhibited the α7 and α3β2 nAChRs, by 55 and 91%, respectively. According to mass spectrometry data, this toxin possessed a molecular mass of 12,358.80 Da. It demonstrated fairly high affinity for α3β2 nAChRs with an IC_50_ value of 566.2 nM. Sequencing by automated Edman degradation resulted in a partial sequence without cysteines: WFRSFKSYYGHHGSVYRPNEPNFRSFAS. This sequence demonstrated similarity, although with low identity, to some turripeptides, containing no cysteines, the sequences of which were deduced from corresponding DNA sequences.

Thus, the number of new nAChR protein ligands discovered over the last decade is quite limited and this leaves ample room for further research.

## 5. Conclusions

Our review shows that the compounds of marine origin in the last decade were in fact extensively used in the studies focusing on the nicotinic acetylcholine receptors. It appears that the most important role is being played by α-conotoxins, peptides from the *Conus* marine snails. It concerns their application in fundamental studies for distinguishing numerous subtypes of the nAChRs in the brain and immune system to get detailed information about their binding sites, identify receptor residues implicated in species selectivity to a particular conotoxin, and thus open new possibilities for drug design. Indeed, there are data that conotoxins have some perspectives as anti-inflammatory and analgesic compounds. The concern of new proteins from marine sources is that they were not widely used in the nAChR research in the last decade. Low molecular weight compounds isolated from marine sources and their numerous synthetic analogs are, as earlier, considered as potential drugs; extensive continued investigation is still required, but the hopes are still on the future.

## Figures and Tables

**Figure 1 biomolecules-12-00189-f001:**
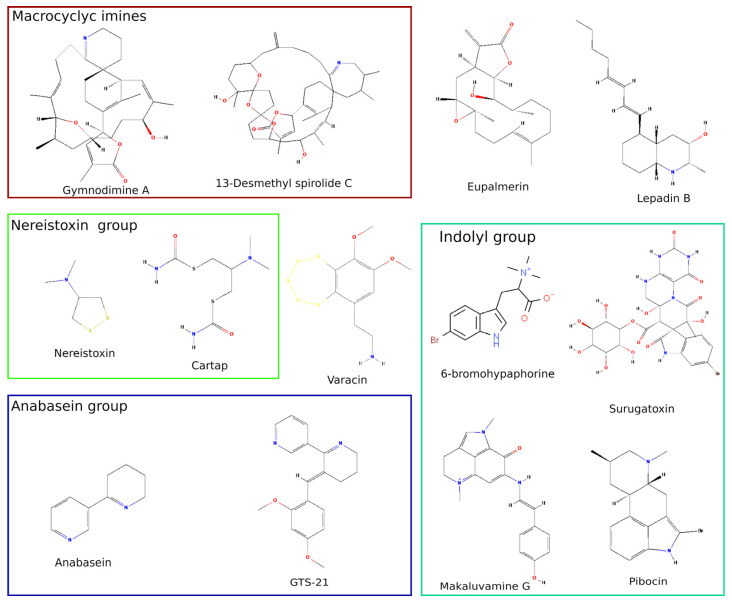
Diversity of low-molecular weight nicotinic acetylcholine receptor (nAChR) ligands of marine origin and some of their synthetic analogs. Nitrogen atoms are indicated in blue, oxygen in red, sulfur atoms in yellow and bromine in brown.

**Figure 2 biomolecules-12-00189-f002:**
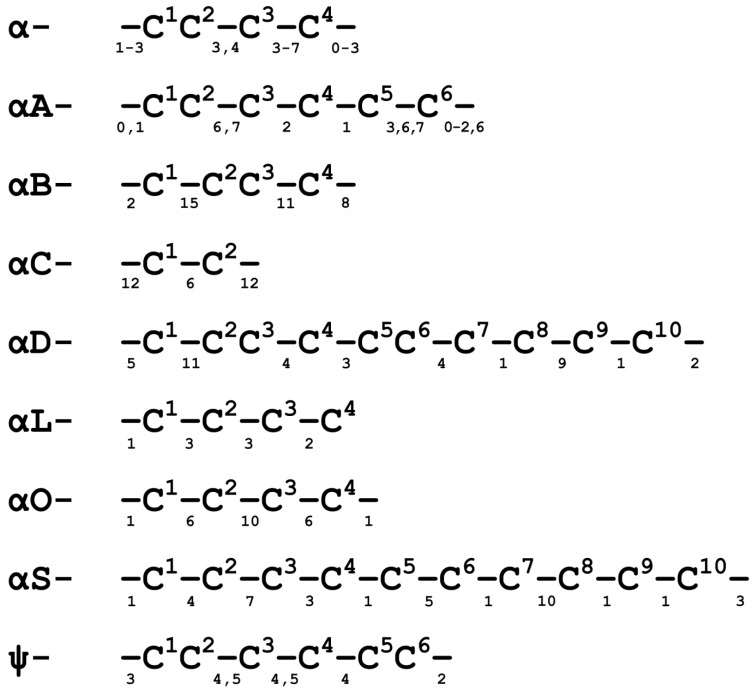
Cysteine framework of the respective conopeptide families targeting nAChRs. The numbers below the dash indicate the number of amino acid residues occurring in native conopeptides between the corresponding cysteine residues.

**Figure 3 biomolecules-12-00189-f003:**
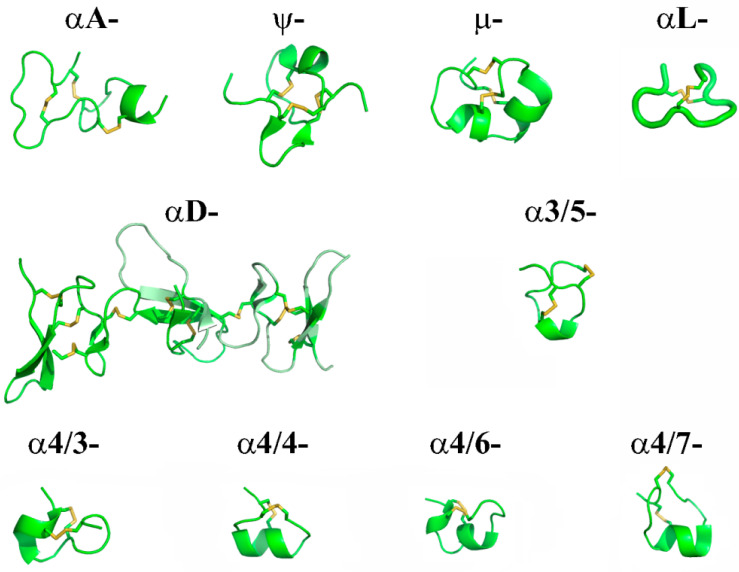
Ribbon visualization of NMR-structures of some representatives of families and groups of conotoxins targeting nAChRs αA-PIVA (PDB ID: 1PIP), ψ-PIIIE (1JLO), μ-CnIIIC (2YEN), αD-GeXXA (4X9Z), αL-lt14a (BMR ID: 21014), α4/3-RgIA (2JUQ), α3/5-CnIA (1B45), α4/4-BuIA (2I28), α4/6-TxID (taken from PDB presentation of molecular-dynamics simulation of the complex between TxID and the binding site of α6β4 nAChR [106]), and α4/7-Bt1.8 (2NAY). Yellow color indicates -S-S-bonds of disulfide bridges.

**Figure 4 biomolecules-12-00189-f004:**
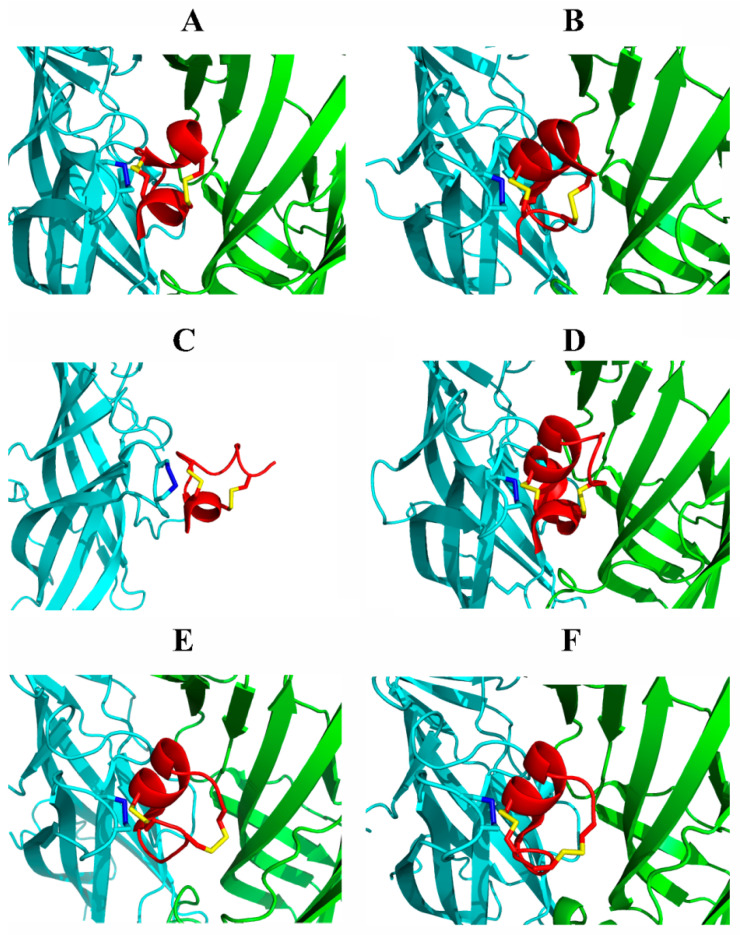
Ribbon representation of the ligand-binding sites in the structures of complexes of some α-conotoxins of different groups with homologues of nAChRs based on the X-ray data. (**A**) α4/3-conotoxin ImI AChBP from *A. californica* (PDB 2C9T), (**B**) α4/4-conotoxin BuIA AChBP from *A. californica* (PDB 4EZ1), (**C**) α4/3-conotoxin RgIA extracellular domain of the α9 subunit of human nAChR (PDB 6HY7), (**D**) α4/7-conotoxin LvIA AChBP from *A. californica* (PDB 5XGL), (**E**,**F**) globular and ribbon isoforms of α5/5-conotoxin AusIA AChBP from *L. stagnalis* (PDB 7N0Y and 7N0W). In all cases (except for the monomeric domain), only two adjacent protomers of AChBP pentamer are represented, the principal side being colored cyan, and the complementary side colored green. All conotoxins are colored red with yellow disulfides. The vicinal disulfide in C-loop of the principal side is colored blue.

**Table 1 biomolecules-12-00189-t001:** Affinity and potency data on marine low molecular weight nicotinic acetylcholine receptor (nAChR) ligands.

Low-Molecular Weight Compound	Target (nAChRs)	IC_50_ ± SEM, or (CI_95%_ ^1^) (nM) Functional Studies	IC_50_ ± SEM (nM) Binding Studies	Ki ± SEM (nM) Binding Studies	Ref.
gymnodimine A	α4β2 (human)	0.5 ± 0.1 ^b^	-	70 ± 19	[30]
	α4β2 (rat)	-	-	68 ± 18	
	α4β4 (human)	8 ± 5 ^b^	-	36 ± 8	
	α6β3β4α5 (human)	-	-	1.0 ± 0.3	
	α7 (human)	2.0 ± 0.1 ^b^	-	1.0 ± 0.1	
	α3*(human)	1.0 ± 0.1 ^b^	-	30 ± 17	
	α3*(rat)	63 ± 2 ^b^	-	8 ± 3	
	(α1)2β1γδ (human)	357 ± 121 ^b^	-	440 ± 215	
	(α1)2β1γδ (mouse)	-	1.38 ± 0.19	-	[21]
	α7-5HT3 (chick)	-	0.33 ± 0.08	-	
	α3β2	-	6.25 ± 1.10	-	
	α4β2	-	15.50 ± 0.19	-	
surugatoxin	α3*(rat superior cervical ganglia)	58	-	-	[26]
13-desmethyl-spirolide C	α4β2 (human)	0.7 ± 0.1 ^b^	-	96 ± 34	[30]
	α4β2 (rat)	-	-	120 ± 15	
	α4β4 (human)	22 ± 4 ^b^	-	43 ± 5	
	α6β3β4α5 (human)	-	-	2.0 ± 0.2	
	α7(human)	0.4 ± 0.1 ^b^	-	0.7 ± 0.2	
	α3*(human)	3.0 ± 0.5 ^b^	-	47 ± 16	
	α3*(rat)	40 ± 1 ^b^	-	24 ± 11	
	(α1)2β1γδ (human)	11 ± 3 ^b^	-	31 ± 6	
20-methyl spirolide G	(α1)2β1γδ (*Torpedo*)	0.36 (0.29−0.45) ^a^	-	0.028 ± 0.005	[20]
	α7(human)	0.48 (0.09–2.50) ^a^	-	-	
	α7-5HT3 (chick)	2.1 (1.4−3.1) ^a^	-	0.11 ± 0.08	
	α4β2 (human)	-	-	3.60 ± 0.07	
	α3β2 (human)	-	-	0.040 ± 0.001	
lepadin B	α4β2 (mouse)	900 (700−1200) ^a^	-	-	[28]
	α7 (mouse)	700 (500−900) ^a^	-	-	
cembranoids	(α1)_2_β1γδ *(Torpedo*)	-	-	435 ± 157	[31]
nereistoxin	α4β2 (chiken)	40,000	-	-	[32]
	α7 (chiken)	33,000	-	-	
	ALS/β2 (Drosophila/chiken)	15,000	-	-	
	SAD/β2 (Drosophila/chiken)	13,000	-	-	
varacin	(α1)2β1εδ (mouse)	8700 ± 400 ^a^	-	-	[13]
	(α1)2β1γδ (*Torpedo*)	-	10,000 ± 1000	-	
	α7 (human)	-	19,000 ± 1000	-	
	*L. stagnalis* AChBP	-	-	790 ± 100	
makaluvamine G	(α1)2β1εδ (mouse)	3300 ± 300 ^a^	-	-	[13]
	(α1)2β1γδ (*Torpedo*)	-	2800 ± 300	-	

^1^ CI95%—95% confidence interval, SEM—standard error of measurement; ^a^ measured from electrophysiology experiments, ^b^ measured from calcium imaging assay; Ref.—reference; α3* means nAChRs of different composition containing the α3 subunit.

**Table 2 biomolecules-12-00189-t002:** New conotoxins published last decade. The italicized names and sequences mean that these peptides were identified by cDNA libraries. The asterisk means the amidated C—terminus, Y—sulfatated tyrosine, O—hydroxyproline, Z—pyroglutamate.

Name	Species	Year	Sequence	Targets			Refs.
				nAChR Subtype	Affinity (nM)	Key Ligand Determinants *	
**α-Conotoxins**							
α-peptide	*C. tinianus*	2011	GGCCSHPACQNNPDYC *	α3β2; α4β2; α7	nd	nd	[114]
*Vc1.2*	*C.**victoriae* (embryos)	2011	*GCCSNPACMVNNPQIC **	α3β2;	75;	N5, A7, N11, N12;	[129]
				α7;	637;	nd;	
				α9α10	~1000	nd	
RegIIA	*C. regius*	2012	GCCSHPACNVNNPHIC *	α3β2;	33;	H14;	[115,130]
				α3β4;	97;	N9, H14;	
				α7	103	N9, N11, N12, H14	
*Mr1.7*	*C* *. marmoreus*	2012	*PECCTHPACHVSHPELC **	α3β2;	53.1;	H13;	[117,118]
				α9α10;	185.7;	E2, S12, H13;	
				α6/α3β2β3	284.2	nd	
*TxIB*	*C. textile*	2013	*GCCSDPPCRNKHPDLC **	α6/α3β2β3	28.4	nd	[119]
LsIA	*C. limpusi*	2013	SGCCSNPACRVNNPNIC *	α3β2;	10;	S1;	[116,131]
				α7;	10;	R10, N12;	
				α3α5β2	31	nd	
EIIA	*C. ermineus*	2013	ZTOGCCWNPACVKNRC *	α1β1γδ	0.46 and 105	nd	[132]
EIIB	*C. ermineus*	2017	ZTOGCCWHPACGKNRC *	α1β1γδ	2.2	nd	[133]
*MrIC*	*C* *. marmoreus*	2014	*PECCTHPACHVSNPELC **	α7	1900	nd	[127,128]
*TxID*	*C. textile*	2013	*GCCSHPVCSAMSPIC* *	α3β4;	3.6–12.5;	G1, H5, P6, V7, M11, P13;	[134,135]
				α6/α3β4	33.9–94.1	G1, H5, P6, V7, S9, M11, P13	
Lo1a	*C. longurionis*	2014	EGCCSNPACRTNHPEVCD *	α7	3240	nd	[124]
BnIA	*C. bandanus*	2014	CCSHPACSVNNPDIC *	α7	~1000	nd	[125]
AusIA	*C. australis*	2014	SCCARNPACRHNHPCV *	α7	10,000–47,000	R5, P7, R10	[107,136]
*LvIA*	*C. lividus*	2014	*GCCSHPACNVDHPEIC* *	α3β2;	8.7–15.6;	G1, H5, P6, N9, H12, P13, I15;	[12,120]
				α3β4;	148–283;	H5, P6, N9, D11, H12;	
				α6/α3β2β3;	108;	nd;	
				α6/α3β4	121	nd	
ViIA	*C. virgo*	2015	RDCCSNPPCAHNNPDC *	α3β2	845.5	H11	[137]
PIC	*C. purpurascens*	2017	SGCCKHPACGKNRC	α1β1ε/γδ; α3β2	nd	nd	[138]
*CIA*	*C. catus*	2018	*NGRCCHPACGKHFSC*	α1β1γδ; α3β2	5.7; 2060	nd	[139]
*CIB*	*C. catus*	2018	*GCCSNPVCHLEHSNLC*	α3β2; α7	128.9; 1510	nd	[139]
*Lt1.3*	*C. litteratus*	2018	*GCCSHPACSGNNPYFC **	α3β2	44.8	N11, N12, P13, F15	[121]
*VnIB*	*C.* *ventricosus*	2019	*GGCCSHPVCYTKNPNCG **	α6β4; α3β4; α6/α3β4	12; 320; 5.3–18	nd	[140]
MilIA	*C.* *milneed-* *wardsi*	2019	DMCCHPACMNHFNC	α1β1ε/γδ	11,000–13,000	M9, N10, H11	[141]
*GIB*	*C. geographus*	2021	*ECCNPACGRHYSCKG* *	α1β1εδ; α9α10	116; 1113	nd	[142]
*G1.5*	*C. geographus*	2021	*GCCSHPACSGNNPEYCRQ**	α3β2; α3β4; α7; α9α10	35.7; 1928; 1935; 569	nd	[142]
*CIC*	*C. catus*	2021	*ASGADTCCSNPACQVQHSDLC*	α3β2; α6/α3β2β3	3510; 1030	nd	[143]
*LvIF*	*C. lividus*	2021	*GCCSHPACAGNNQDIC **	α3β2; α6/α3β2β3	9.0; 14.4	nd	[122]
*Bt1.8*	*C. betulinus*	2021	*GCCSNPACILNNPNQC **	α6/α3β2β3; α3β2	2.1; 9.4	I9, N11, N12; I9, L10, N11, N12, N14, Q15	[123]
**αL-Conotoxins**							
*lt14a*	*C. litteratus*	2006	*MCPPLCKPSCTNC **	neuronal nAChRs	nd	nd	[144,145]
*Pu14a*	*C. pulicarius*	2010	*DCPPHPVPGMHKCVCLKTC*	α1β1γδ; α6α3β2	<1000; ~1000	nd	[146]
**αO-Conotoxins**							
*GeXIVA*	*C. generalis*	2015	*TCRSSGRYCRSPYDRRRRYCRRITDACV* *	α9α10	4.61	nd	[100]
GeXXVIIA	*C. generalis*	2017	ALMSTGTNYRLLKTCRGSGRYCRSPYDCRRRYCRRISDACV	α9α10; α1β1εδ	16.2; 774	C-terminal part (27–41); N-terminal part (1–26)	[147]

* Amino acid residues were recognized as key determinants if, when they were removed or replaced, the affinity for the corresponding target decreased by more than five times; nd—not determined.

## Data Availability

Not applicable.

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
