# Peer review of "Marine Origin Ligands of Nicotinic Receptors: Low Molecular Compounds, Peptides and Proteins for Fundamental Research and Practical Applications"

_biomolecules, 2022, doi:10.3390/biom12020189_

Round 1

Reviewer 1 Report

The review is well thought through. It gives a comprehensive overview of various compounds of marine origin, from low molecular weight to proteins, that act at nicotinic acetylcholine receptors (nAChRs). However, a brief description of structural determinants required for activity at nAChRs for each group of compounds would be beneficial to readers. It would alleviate searches in metabolomic, transcriptomic and proteomic databases for potential ligands of nAChRs. Also, a little rehearsal of where and why linear and globular forms of peptides and proteins come from would be helpful.

Overall, the review is of uneven quality. Although chapters 4 and 5 are immaculate and chapters 1 and 2 need only minor typographic and linguistic corrections (e.g., to-day -> today or to date, mutating -> mutation, releasing -> yielding, specific -> selective, admitted -> introduced, ...), chapter 3 needs substantial editing. I am not a native speaker but grammatical errors are eye-striking. At many places, the subject is missing (e.g., ln 361), the sentence has a redundant predicate (e.g., ln 326-329, ln 363-365), or the predicate precedes the subject (e.g., ln 406-407, ln 416-417). Moreover, articles are either missing or misused (e.g., ln 394 should read “the new representative”). Please double-check that you state what you mean (e.g., “a few” = several vs “few” = only a little bit). Chapter 3 contains a lot of hard-to-understand compound sentences (e.g., ln 316-320, ln 326-329, ln 468-471, ln 501-504). I would suggest splitting them to improve readability. Moreover, pharmacological terms are misused (e.g. potency and affinity are mixed up at ln 355, ln 448-452).

Ln 489-491, The statement should be specified in what “information” says.
Ln 509 “bases” -> “basis”
Ln 509 “species selectivity” -> “specie selectivity” or “selectivity among species”
Ln 524 “different” -> “various”
Ln 558-559 “was established in the work [171].” What work? Work of whom?
Ln 621-626 It is incomprehensible what authors try to say.

Author Response

However, a brief description of structural determinants required for activity at nAChRs for each group of compounds would be beneficial to readers. It would alleviate searches in metabolomic, transcriptomic and proteomic databases for potential ligands of nAChRs.

We agree with this proposal, but a large amount of data on the key determinants of marine origin ligands for their cholinergic activity would require a separate review. Please, note that similar aspects concerning conotoxins are covered in our review (references [175, 176]). In order to follow the specified purpose (“to alleviate searches in transcriptomic and proteomic databases for potential ligands of nAChRs”), it would be necessary to collect the huge amount of relevant data available to date. However, in this review we concentrated on information for the last 10 years, and, accordingly, a large array of data on key determinants of ligands studied earlier fell out of our consideration (this is especially true for low-molecular compounds and proteins). Nevertheless, in order to partially answer this question, we have collected relevant data on conotoxins discovered and studied over the past decade focusing on the key residues responsible for their affinity for a distinct target (the collected information and references are given in the updated Table 2).

Also, a little rehearsal of where and why linear and globular forms of peptides and proteins come from would be helpful.

We originally planned to limit our review to the consideration of mainly naturally-occurring compounds, avoiding, if possible, the inclusion of works investigating only analogues, including molecules with non-natural folding. Nevertheless, answering the question, we have added a small comment on the reasons for the synthesis and structural and functional studies of such compounds (lines 306-313).

Overall, the review is of uneven quality. Although chapters 4 and 5 are immaculate and chapters 1 and 2 need only minor typographic and linguistic corrections (e.g., to-day -> today or to date, mutating -> mutation, releasing -> yielding, specific -> selective, admitted -> introduced, ...), chapter 3 needs substantial editing. I am not a native speaker but grammatical errors are eye-striking. At many places, the subject is missing (e.g., ln 361), the sentence has a redundant predicate (e.g., ln 326-329, ln 363-365), or the predicate precedes the subject (e.g., ln 406-407, ln 416-417). Moreover, articles are either missing or misused (e.g., ln 394 should read “the new representative”). Please double-check that you state what you mean (e.g., “a few” = several vs “few” = only a little bit). Chapter 3 contains a lot of hard-to-understand compound sentences (e.g., ln 316-320, ln 326-329, ln 468-471, ln 501-504). I would suggest splitting them to improve readability.

We are very grateful for your corrections and also apologize for misprints. All the indicated errors were corrected. In addition, we have once again carefully edited the entire text and especially Chapter 3, trying to simplify the phrases.

Moreover, pharmacological terms are misused (e.g. potency and affinity are mixed up at ln 355, ln 448-452).

It was corrected.

Ln 489-491, The statement should be specified in what “information” says.

The whole paragraph has been rewritten (lines 510-527).

Ln 509 “bases” -> “basis”
Ln 509 “species selectivity” -> “specie selectivity” or “selectivity among species”
Ln 524 “different” -> “various”
Ln 558-559 “was established in the work [171].” What work? Work of whom?

We have corrected all the noted errors.

Ln 621-626 It is incomprehensible what authors try to say.

The whole paragraph has been rewritten (lines 653-659).

Reviewer 2 Report

The work of Kasheverov et al. focuses on reviewing the marine ligands targeting nAChR. The manuscript overall does a good job in explaining the structural and functional properties of these ligands, but there are minor changes that need to be done to improve the manuscript. There are many linguistic errors in the manuscript that require professional proofreading. As for the scientific changes, below are my suggestions:

  • P2 line 51, there should be another work: https://pubmed.ncbi.nlm.nih.gov/33958730/
  • P7 line 229, there are other compounds based on anabaseines that show allosteric effects that should be included, https://doi.org/10.1016/j.bmcl.2013.05.039 is one example.
  • P11 line 365, the right term would be a “silent agonist” DOI: 1124/jpet.114.215236.
  • Chapter 3.3, Abba Leffler has several papers on this topic, which should be included.
  • Chapter 3.4 merits a discussion of conotoxin derivatives with non-canonical amino acids that are likely to have better therapeutic properties than regular peptides.
  • Chapter 4.1 would benefit from the discussion of the results from https://doi.org/10.3390/toxins12090598.

Author Response

The work of Kasheverov et al. focuses on reviewing the marine ligands targeting nAChR. The manuscript overall does a good job in explaining the structural and functional properties of these ligands, but there are minor changes that need to be done to improve the manuscript. There are many linguistic errors in the manuscript that require professional proofreading.

We are grateful for good words about our review, apologize for the grammar errors and misprints and tried to get rid of them in the revised version of the manuscript.

  • P2 line 51, there should be another work: https://pubmed.ncbi.nlm.nih.gov/33958730/
  • We have added this reference to the text, but in another more appropriate place (reference [180], line 607 in the last paragraph of Chapter 3.3).

  • P7 line 229, there are other compounds based on anabaseines that show allosteric effects that should be included, https://doi.org/10.1016/j.bmcl.2013.05.039 is one example.
  • The indicated publication is now in the References ([91], line 236).

  • P11 line 365, the right term would be a “silent agonist” DOI: 1124/jpet.114.215236.
  • It was corrected.

  • Chapter 3.3, Abba Leffler has several papers on this topic, which should be included.
  • Two publications by Abba Leffler are introduced and discussed in Chapter 3.3 ([154,155], lines 503-508).

  • Chapter 3.4 merits a discussion of conotoxin derivatives with non-canonical amino acids that are likely to have better therapeutic properties than regular peptides.
  • Due to a large volume of available studies on conotoxins concerning the stated topic, we initially decided to limit our review to the publications of the last decade and planned to avoid non-natural analogues. In addition, the therapeutic use of the described compounds was not considered as the main part of this work and is therefore presented as brief auxiliary parts. Of course, the role of non-canonical amino acid residues for the functional activity and clinical prospects of conotoxins is very interesting, but this will need a selection of relevant literature and most probably preparing a separate review. As a compensation, we included a brief mentioning of this aspect in Chapter 3.4 (lines 637-639) and supplemented it with a reference [187], in which there is a small section on analogues of conotoxins with non-canonical amino acid residues.

  • Chapter 4.1 would benefit from the discussion of the results from https://doi.org/10.3390/toxins12090598.
  • We are grateful for this suggestion and the discussion of the results from Gulsevin & Meiler paper was added to the manuscript (lines 670-678 with additional references [211,212]).